# The Impact of Antimicrobial Stewardship and Infection Control Interventions on *Acinetobacter baumannii* Resistance Rates in the ICU of a Tertiary Care Center in Lebanon

**DOI:** 10.3390/antibiotics11070911

**Published:** 2022-07-07

**Authors:** Nesrine A. Rizk, Nada Zahreddine, Nisrine Haddad, Rihab Ahmadieh, Audra Hannun, Souad Bou Harb, Sara F. Haddad, Rony M. Zeenny, Souha S. Kanj

**Affiliations:** 1Division of Infectious Diseases, Department of Internal Medicine, American University of Beirut Medical Center, Beirut 1107 2020, Lebanon; nr00@aub.edu.lb (N.A.R.); sb125@aub.edu.lb (S.B.H.); sarahaddad711@gmail.com (S.F.H.); 2Infection Control and Prevention Program, American University of Beirut Medical Center, Beirut 1107 2020, Lebanon; nk13@aub.edu.lb (N.Z.); ra255@aub.edu.lb (R.A.); 3Department of Pharmacy, American University of Beirut Medical Center, Beirut 1107 2020, Lebanon; nh126@aub.edu.lb (N.H.); audrahannun@gmail.com (A.H.); rz37@aub.edu.lb (R.M.Z.)

**Keywords:** *Acinetobacter*, carbapenem-resistant *A. baumannii* (CRAb), infection control, antimicrobial agents, carbapenems, antibiotic resistance, clinical pharmacy services, antimicrobial stewardship, intensive care

## Abstract

Antimicrobial resistance is a serious threat to global health, causing increased mortality and morbidity especially among critically ill patients. This toll is expected to rise following the COVID-19 pandemic. Carbapenem-resistant *Acinetobacter baumannii* (CRAb) is among the Gram-negative pathogens leading antimicrobial resistance globally; it is listed as a critical priority pathogen by the WHO and is implicated in hospital-acquired infections and outbreaks, particularly in critically ill patients. Recent reports from Lebanon describe increasing rates of infection with CRAb, hence the need to develop concerted interventions to control its spread. We set to describe the impact of combining antimicrobial stewardship and infection control measures on resistance rates and colonization pressure of CRAb in the intensive care units of a tertiary care center in Lebanon before the COVID-19 pandemic. The antimicrobial stewardship program introduced a carbapenem-sparing initiative in April 2019. During the same period, infection control interventions involved focused screening, monitoring, and tracking of CRAb, as well as compliance with specific measures. From January 2018 to January 2020, we report a statistically significant decrease in carbapenem consumption and a decrease in resistance rates of isolated *A. baumannii*. The colonization pressure of CRAb also decreased significantly, reaching record low levels at the end of the intervention period. The results indicate that a multidisciplinary approach and combined interventions between the stewardship and infection control teams can lead to a sustained reduction in resistance rates and CRAb spread in ICUs.

## 1. Introduction

Antimicrobial resistance was recognized as a serious threat to global health several years before the onset of the COVID-19 pandemic [1]. In fact, a report published in 2016 by the World Bank and the World Health Organization (WHO) predicted that antimicrobial resistance could lead to 10 million deaths each year by 2050 [2,3] while a more recent study estimated that, globally, 4.95 million deaths were associated with resistant bacteria in 2019 [4]. In addition to the resulting mortality, the increase in morbidity, disability and hospital length of stay lead to increased costs with direct negative consequences on the global economy [5,6]. Antimicrobial resistance is a major concern for the developing world, with economic- and public health-related repercussions especially due to the spread of resistant Gram-negative pathogens [7,8] that are leading multidrug resistance around the world. Those organisms feature on the critical priority pathogens list of the WHO [9,10,11]. They are associated with nosocomial infections, specifically in acute care settings and intensive care units (ICUs) [12]. A recent report from the WHO Eastern Mediterranean region revealed alarming rates of multidrug-resistant pathogens including carbapenem-resistant *Acinetobacter baumannii* (CRAb) which is the most common pathogen in Gram-negative bacteremia [13]. As a response to the antimicrobial resistance threat, the WHO launched in 2015 a Global Action Plan against antimicrobial resistance comprising multiple interventions based on five objectives [14]. Among those, Antimicrobial Stewardship and Infection Control are important strategies that aim to guide the judicious use of antimicrobials and control the spread of resistant microorganisms within healthcare institutions [15].

Even prior to the COVID-19 pandemic, antimicrobial misuse and overuse in critical care settings was very common with the frequent utilization of multiple broad-spectrum antibiotics for long courses of therapy [12]. Several risk factors put critically ill patients at higher risk of colonization and infection with multidrug-resistant organisms including treatment with immunosuppressive drugs, use of invasive devices, exposure to a wide range of antibiotics, and prolonged hospitalizations [16]. Following the COVID-19 pandemic, resistance rates are expected to increase [17,18] as COVID-19 has led to an influx of critically ill patients who often receive unnecessary antibiotic therapy [19,20]. A report by the Center for Disease Control published in February 2021 described outbreaks of antimicrobial resistant infections in COVID-19 units such as CRAb and *Candida auris* (*C. auris*) [19] with a noticeable increase in the overall hospital-acquired infections, most of which are caused by multidrug-resistant organisms [21]. On the other hand, the pandemic may have a positive impact on antimicrobial resistance as there may be a possible decrease in the transmission of resistant organisms, as a direct consequence of global travel restrictions, more frequent hand hygiene, social distancing, as well as enhanced infection control practices globally [22].

CRAb is among the most resistant organisms of the *Acinetobacter* species. It is ubiquitous in nature and in addition to its resistance to carbapenems, it is intrinsically resistant to a large number of antimicrobial agents and has the potential to develop additional resistance and cause infections in humans [23]. *A. baumannii* is unique in that it possesses an excellent genome plasticity; it has the ability to take any gene from its surroundings. This feature might have played a crucial role in the evolution of this human opportunistic pathogen towards clinical success and being a multidrug-resistant pathogen [24]. It has an island of drug-resistant genes in its genome that makes it different from other superbugs [25]. *A. baumannii* is the most prevalent carbapenem-resistant organism worldwide [26] and is associated with hospital-acquired infections causing a significant increase in morbidity and mortality [27] especially in patients admitted to ICUs [23,28]. In the East Mediterranean region, CRAb is notoriously implicated in major outbreaks in healthcare settings [29]. During the last decade, wars and violent conflicts have contributed to the spread of this organism from combat areas to hospitals treating the war-injured and refugees [30,31]. The detrimental impact of antimicrobial resistance and CRAb on public health was recognized in this region, prompting governments and experts to collaborate under the WHO umbrella to tackle antimicrobial resistance [32] and develop recommendations for the treatment of CRAb and other multidrug resistant organisms [33]. 

CRAb is responsible for most of the severe infections in ICUs worldwide [34] in patients colonized or infected with it. CRAb is defined as any *A. baumannii* isolate that is resistant to carbapenems. These isolates are usually also resistant to most antibiotics excluding polymyxin E (colistin) and tigecycline. As early as 1980, and following armed clashes during the Lebanese civil war, an increase in CRAb was reported from our hospital [35]. A recent review on carbapenem resistance among *A. baumannii* isolates revealed increasing resistance rates in Lebanon [36]. In fact, *A. baumannii* comprised 82% of isolates collected from 16 Lebanese hospitals in the years 2011–2013 [37] and 87% among samples from 13 Lebanese hospitals [38] in the years 2015–2016. Other reports from major Lebanese hospitals reveal the burden of CRAb on ICUs, with increased mortality and morbidity and poor patient outcomes [39,40,41,42]. Interventions to control CRAb in those hospitals included either infection control measures to break transmission—such as terminal cleaning of an ICU [43]—or antimicrobial stewardship efforts to decrease resistance rates [44,45]. *Acinetobacter baumannii* constitute the large majority of the *Acinetobacter* organisms tested in our microbiology diagnostic laboratory. For the purpose of this study, all *Acinetobacter* species will be referred to as *Acinetobacter baumannii* [46,47].

Similar to the other medical centers in the region, we struggle with high rates of resistance among Gram-negative bacteria, mainly the extended spectrum beta-lactamase-producing (ESBL) Enterobacterales. Therefore, carbapenem use is widespread [29]. Carbapenem consumption has been found to be associated with increasing rates of CRAb [48]. CRAb is a pathogen of concern in our hospital, where according to targeted surveillance efforts, the rates of CRAb sharply increased from 52% in 2010 to peak at 92% in 2012 [49]. A prospective study conducted at our center between 2007 and 2014 showed that the most common site for isolating CRAb was the respiratory tract, notably in patients with ventilator-associated pneumonia (VAP) [39]. CRAb was also the predominant pathogen, both in early- and late-onset VAP, in a retrospective review on VAP published in 2019 [50]. The pattern of resistance of CRAb at AUBMC is quite similar to those reported from neighboring Arab countries, with the predominance of the blaOXA-23 gene. *A. baumannii* isolated from our hospital tend to be multidrug resistant (to trimethoprim–sulfamethoxazole, quinolones, aminoglycosides, and beta-lactam antibiotics) [46]. 

Accordingly, we find it essential to develop concerted interventions to control the spread of CRAb. In our study, we describe the impact of combined antimicrobial stewardship and infection control interventions on resistance rates of *Acinetobacter baumannii* and colonization pressure of CRAb in our ICU prior to the onset of COVID-19 pandemic. 

## 2. Materials and Methods

### 2.1. Hospital Setting

The American University of Beirut Medical Center (AUBMC) is a leading tertiary care medical center (364 beds) serving patients from Lebanon and neighboring countries. Its medical and surgical services are the busiest in the nation with a medical and surgical ICU comprising 30 single-bed rooms. The adult ICU population at the AUBMC consists of high-risk patients with multiple comorbidities, immunocompromised patients, trauma patients, as well as patients following major surgical procedures. AUBMC ICU also receives referred patients from other facilities in the country as well as from Syria and Iraq, countries inflicted by war. In November 2018, the AUBMC acquired the EPIC electronic medical record software [51]. EPIC is a cloud-based electronic health record software built for hospitals. The transition to a fully automated health medical record allowed for additional opportunities for antimicrobial monitoring and targeted infection control interventions.

### 2.2. Antimicrobial Stewardship

Actions led by antimicrobial stewardship programs are essential to control the misuse and abuse of antimicrobials and decrease healthcare costs and antimicrobial resistance [52,53,54]. Antimicrobial stewardship efforts started at AUBMC in 2007. However, the antimicrobial stewardship program was formally launched in June 2018, with a dedicated team composed of an Infectious Disease physician and a pharmacist [55]. The objectives of the antimicrobial stewardship program are to optimize patient safety, reduce the emergence of antimicrobial resistance and decrease hospitalization costs [54,56,57]. The stewardship team reviews patients’ antimicrobial therapies daily and provides prospective audits and feedback on the use of broad-spectrum antibiotics in addition to calculating and reporting overall antimicrobial consumption, developing and implementing guidelines to standardize and optimize antimicrobial use at the institution, and finally offering ongoing educational activities to healthcare providers. 

#### 2.2.1. Antimicrobial Stewardship Interventions

Due to the emergence of carbapenem resistance, namely among *Acinetobacter baumannii*, the antimicrobial stewardship team introduced, in April 2019, an initiative for carbapenem sparing with the aim of reducing carbapenem consumption and assessing the impact on *Acinetobacter baumannii* carbapenem resistance rates. Even with carbapenems being the mainstay of therapy for ESBL-producing organisms, recent data and guidance suggest using alternatives to carbapenems in several scenarios (intra-abdominal infections, complicated urinary tract infections and pyelonephritis, oral step-down therapy, and surgical prophylaxis) to try to limit carbapenem use. We implemented a carbapenem-sparing approach focused on the intensive care units during this month [55]. As such, the stewardship team conducted daily stewardship handshake rounds and reviewed the charts of all ICU patients receiving carbapenems. The stewardship team assessed the appropriateness of carbapenem use (appropriate/not appropriate) (opinion of the infectious diseases specialist and pharmacist after chart review). The non-appropriate prescriptions of carbapenems were categorized as follows: duration of therapy, dose adjustment, de-escalation, duplicate coverage, drug–bug mismatch, IV to oral switch. The stewardship team proposed alternatives to the inappropriate carbapenem prescriptions when applicable; those were labeled as “interventions”. At the end of this month, we calculated the rate of acceptance of those interventions (accepted/not accepted) and compared the acceptance rates at the beginning versus acceptance rates at the end of the intervention month. Stewardship rounds were coupled with didactic lectures on principles and applications of antimicrobial stewardship to medical interns, residents, infectious diseases fellows, and pharmacists. At the end of this project, the stewardship team resumed their daily operations as described above.

#### 2.2.2. Antimicrobial Stewardship Measures

To assess the impact of the carbapenem-sparing strategy, we adopted the following quantitative metrics to measure carbapenem antibiotic consumption: defined daily dose (DDD) and days of therapy (DOT). Quantitative metrics were calculated at baseline, before the initiative implementation and, subsequently on a monthly and quarterly basis, after implementation [58]. Table 1 illustrates the formulas used to calculate DDD and DOT [58,59,60] on a quarterly and monthly basis respectively.

### 2.3. Infection Control 

The Infection Control and Prevention Program was established at AUBMC in 1980. Infection control strategies have included surveillance, prevention and management of outbreaks, environmental hygiene, and optimization of employee health and education [56]. The Infection Control team at the AUBMC tracks multidrug-resistant organisms in the hospital. Reports for Methicillin-resistant *Staphylococcus aureus* (MRSA), Vancomycin-resistant enterococci (VRE), Carbapenem-resistant Enterobacterales (CRE), multidrug-resistant *A. baumannii*, difficult to treat *Pseudomonas aeruginosa*, and, more recently, *Candida auris* are generated on daily basis. Clusters and outbreaks are closely monitored and investigated especially in critical care units. During the last decade, several CRAb clusters and outbreaks were identified in our ICUs [33]. The infection control team recognized this threat and implemented an active surveillance for CRAb for all ICU admissions to detect colonization or infections: ICU patients are screened for CRAb upon admission and placed on contact isolation pending the culture results. Moreover, the results of the clinical cultures obtained during the patient’s stay in ICU are analyzed to differentiate hospital-acquired transmissions from community-acquired infections or colonization with CRAb.

Multiple interventions were introduced by the infection control team throughout the years as part of an intensified effort to curb the spread of CRAb. Screening all ICU admissions was one of the major interventions to detect the carriage of CRAb and other carbapenem-resistant organisms. A screening method was adopted for CRAb and CRE, to collect swabs from the oropharynx, bilateral axilla, umbilical and perianal areas as well as from the rectum. Moreover, all patients admitted to ICU were bathed using Chlorhexidine 4% solutions to decrease the bacterial load on their skin and reduce bacterial transmissions [61]. Furthermore, infection control prevention bundles (ventilator bundle, urinary catheter bundle, and central line bundle) were adopted to improve the processes for care of patients. Certifications for the insertion and care of central lines became mandatory for the medical and nursing teams, and are granted after taking an online course. Several practices were also introduced to reduce environmental contamination outside of the ICU. Practices such as restricting the transport of patients unless urgently needed, cleaning and disinfection of the elevators used and CT premises after imaging, or any other visited area, are used.

Staff education and training on hand hygiene and principles of nosocomial transmission of multidrug-resistant organisms were conducted monthly. Each session included all infection control breaches and observations to improve staff practices in ICU. During these sessions, feedback reports and identified breaches were presented, and opportunities for improvement were discussed. Training on hand hygiene included all five-evidence based key moments as per the WHO recommendations [62]. Alcohol hand rubs, at a concentration of 70% ethanol or propanol, were installed at the door of each patient’s room. Compliance was closely monitored with the assistance of anonymous auditors, and feedback reports were regularly communicated to managers and hospital leadership. Closed-circuit television (CCTV) surveillance cameras were installed in three critical care units in 2015. All noted breaches from live and retrospective reviews are promptly reported to nurse managers of the unit for appropriate action. The infection control team conducted intensified rounds to observe practices, raise awareness and improve compliance of the ICU staff with all needed measures. Tiered hand hygiene accountability interventions were adopted based upon a validated model [63] and this was reflected in the hospital hand hygiene policy. Interventions started with direct feedback followed by the awareness intervention, then the authority intervention and ending with the disciplinary intervention. Hand hygiene compliance rates started to improve for the physician group as a result. Hand hygiene compliance rates were sustained and improved further at the start of the COVID-19 pandemic. In addition, visitors were restricted to decrease environmental contamination as per a new visitation policy. An important measure was also added, where nurses were assigned to monitor healthcare workers and visitors during the day shifts; their role was to promptly intervene whenever infection control breaches were observed [50].

The direct patient environment plays a major role in transmitting multidrug-resistant pathogens among patients. Contaminated surfaces contribute to CRAb transmission to vulnerable patients. Routine environmental cultures to identify sources of environmental contamination with CRAb (mattresses, pillows, keyboards) were introduced. After each patient discharge, manual cleaning/disinfection was conducted followed by air decontamination using hydrogen peroxide (H_2_O_2_) at a concentration of 1% (generating 4.7% boosted H_2_O_2_). Environmental cultures that were taken initially were discontinued following sustained negative culture results of the patient environment. Obtaining new cleaning and disinfection solutions and changes in housekeeping processes were also instrumental in improving the patient care environment. All the changes were reflected in updated policies and were reinforced through structural staff training. 

The carriage on admission and acquisition during ICU stay of CRAb was calculated using the CRAb colonization pressure (Table 1). Colonization pressure is defined as the proportion of patients colonized with CRAb in an ICU during a specific period. It reflects the burden of CRAb in an ICU and can estimate the probability of CRAb transmission in this setting. Thus, any new transmission (colonization or infection) of CRAb is strongly correlated to colonization pressure.

Resistance to carbapenems among *Acinetobacter baumannii* at our hospital was the main outcome of this study. Carbapenem resistance among *Acinetobacter baumannii* is routinely reported by our microbiology laboratory. *Acinetobacter* isolates were identified using the Matrix-Assisted Laser Desorption Ionization (MALDI-TOF) Time-of-Flight Mass Spectrometry (MALDI-TOF) platform, and all isolates were tested using the disk diffusion method based on the Clinical and Laboratory Standards Institute (CLSI) breakpoints. We relied on resistance rates reported by the laboratory to follow the outcome of our interventions on resistance rates. 

## 3. Results

### 3.1. Antimicrobial Stewardship Results

The antimicrobial stewardship team launched its daily operations in January 2019 and collected data on the appropriateness of broad-spectrum antibiotic use across the hospital. Those recommendations were labeled as “stewardship interventions”. Our focused intervention in the ICU (the carbapenem sparing strategy) started in April 2019 and yielded the following results over a one-month period: among patients who were prescribed broad-spectrum antibiotics, 188 patients (or 14.6% of the ICU patients during this month) were receiving carbapenem therapy. A total of 81 interventions were recorded during this month in adult patients and included the de-escalation of therapy (23%), dose change (28%) and limiting the duration of therapy (23%). Therefore, combined recommendations to discontinue carbapenem therapy (de-escalation or stop) comprised 46% of all interventions as shown in Figure 1. The overall acceptance rate of recommendations during this intervention period (April 2019) was 73%. As a result of all antimicrobial stewardship efforts, for 2019, there was an increase in stewardship interventions’ acceptance rate from 16.66 to 55.95% when comparing January 2019 to January 2020 (*p* = 0.03). Even though the antimicrobial stewardship team was active, the efforts were less focused and spanned over the whole hospital (vs. April 2019, when the efforts were focused on the ICUs).

In analyzing the indication for use of carbapenems by the antimicrobial stewardship team, we defined an appropriate empiric therapy with carbapenem as follows: patient is a candidate for broad antibiotic therapy and warrants carbapenem usage such as recent culture with ESBL Enterobacterales or other multidrug-resistant organisms, sepsis, or febrile neutropenia. Therefore, 88% of empiric carbapenem prescriptions were deemed appropriate initially and may have required subsequent adjustment based on culture results. As such, the antimicrobial stewardship team found that indication for use, dosing, and duration were appropriate in 88, 80, and 89% of the cases, respectively (Figure 2). As part of our analysis of those results, when comparing the months of January 2019 and January 2020 pre- and post-intervention period, the indication for use in empirical therapy before 48 h changed from 86.4 to 92.9%. Similarly, indication for use in empirical therapy after 48 h from culture results, and indication for targeted therapy was appropriate in 67.1% (January 2019) and 78.9% (January 2020) of cases, and 88.9% (January 2019) and 91.4% (January 2020) of cases, respectively. Duration and dosing regimens were appropriate in 64.3 and 75.8% of cases in January 2019, respectively, as opposed to appropriateness rates of 72.1 and 68.7% in January 2020. 

Additional measures such as infection with *Clostridium difficile* rates, hospitalization costs, and the impact of our interventions and recommendations on patient outcomes were not studied during this time.

The overall carbapenem consumption across the hospital was reflected by carbapenem DOT and DDD, with the greatest volume of consumption occurring in the critical care units. Figure 3 demonstrates the decrease in carbapenem DDD since 2018 and until December 2020. Both DOT (shown later in the text) and DDD trends show a decrease in the consumption that is better seen starting in the second quarter (Q2) of 2019 with the intensification of the carbapenem-sparing efforts. This decrease was maintained in 2019, however, there is a noticeable increase in both DDD and DOT in 2020 compared to 2019, albeit the carbapenem consumption was still lower than 2018.

### 3.2. Infection Control Results

Following the implementation of the intensive infection control measures listed above there was a noticeable improvement in compliance with measures (such as hand hygiene) and reduced colonization pressure of CRAb.

Compliance with hand hygiene is associated with positive patient outcomes. The prevalence of hospital acquired infections was reduced by more than 40% at Geneva University Hospital when compliance rate increased from 48 to 66% over a 5-year period [62]. Figure 4 shows results at our center with improved compliance from 74% to more than 95%. 

A sustained improvement of infection control practices was noticed across the hospital and especially in the ICU. This was reflected in the persistent decrease in the CRAb colonization pressure over the years as shown in Figure 5. CRAb transmission rates in ICU decreased steadily: *A. baumannii* colonization pressure was 340 per 1000 patient days in 2015, 221 per 1000 in 2016, 218 per 1000 in 2017 and 112.7 per 1000 in 2018. The colonization pressure decrease in 2019 became evident and reached 18.4 per 1000 during the second quarter of 2019.

The carbapenem-sparing strategy, combined with the infection control interventions, led to a significant decrease in CRAb colonization pressure rates among ICU patients. Figure 6 shows the colonization pressure per quarter in relation to carbapenem consumption reflected by the carbapenem DOT for 2018 to 2020. The sustained decrease in CRAb transmissions (infections and colonization) is more clearly seen in Figure 6 where the colonization pressure for CRAb is correlated with DOT per quarter from 2018 to 2020, following the beginning of stewardship efforts in 2019 and the launch of the carbapenem-sparing strategy in April 2019. Colonization pressure rates decreased steadily from 210.4 per 1000 patient days in Q4-2018 to 0 per 1000 patient days in Q1-2020.

The major finding in our study was the impact on carbapenem resistance rates among *Acinetobacter*
*baumannii* in our institution (Figure 7). The continuous monitoring of resistance rates allows the antimicrobial stewardship and infection control teams to measure the ongoing and long-term effects of their interventions. Figure 7 shows the rates of resistance to carbapenems among the *Acinetobacter*
*baumannii* at the AUBMC over a decade and highlights the continuous but slow decline in resistance rates since 2014, followed by a sharp drop in 2020. The rates of carbapenem resistance among collected CRAb at the AUBMC peaked at 92% in 2012 and were slowly declining with the intensification of infection control measures and some antimicrobial stewardship efforts. However, only following the implementation of the antimicrobial stewardship program at the end of 2018 did the resistance rates among CRAb decreased in 2020 to 63%.

## 4. Discussion

The containment of CRAb is difficult to achieve in acute care settings; however it is expected to result in a significant reduction of mortality and morbidity especially among critically ill patients [64]. Carbapenem consumption is linked to increasing *Acinetobacter baumannii* resistance rates while nosocomial transmission is linked to environmental contamination, invasive procedures, and patient vulnerabilities [64,65]. 

Mathematical models have described the potential impact of reducing carbapenem consumption on resistance acquisition among bacteria, including CRAb [66], and antimicrobial stewardship to restrict carbapenem usage has been suggested for controlling outbreaks caused by CRAb in critical care settings [67,68,69]. There is an abundance of reports and studies on the effectiveness of infection control measures in limiting the transmission of CRAb in hospital ICUs. Environmental cleaning appears to be particularly important [70,71] as well as enforcing strict hand hygiene compliance policies among healthcare workers [72]. Compliance with hand hygiene is associated with positive patient outcomes; the prevalence of hospital acquired infections was reduced by more than 40% at Geneva University Hospital when compliance rate increased from 48 to 66% over a 5-year period [62]. Very few reports describe the impact of combined infection control and antimicrobial stewardship interventions on colonization pressure and resistance rates among CRAb isolates [73,74]. Our results indicate that a multidisciplinary approach and conjoined efforts of antimicrobial stewardship and infection control teams can lead to a sustained reduction in CRAb spread in the ICU. 

One main finding of our study relates to the decrease in resistance rates among *Acinetobacter baumannii* to imipenem from 81% in 2018 to 63% in 2020. *A. baumannii* accounts for 99% of *Acinetobacter baumannii* in our hospital. Therefore, this reduction is significant, and reflective of the effectiveness of a carbapenem-sparing strategy at the level of the hospital with an intensification of daily stewardship interventions especially in the ICU and continued educational efforts. A significant reduction in CRAb colonization pressure was demonstrated as a 200-fold decrease during the two-year study period. The decrease in CRAb colonization pressure over the years was mainly the result of ongoing infection control interventions following the identification of each cluster or outbreak of CRAb. The launching of the carbapenem-sparing strategy by the antimicrobial stewardship team during the second quarter of 2019 led to a sustained decrease in colonization pressure over subsequent quarters as shown in Figure 6. Strict antimicrobial stewardship combined with comprehensive infection control measures resulted in successfully controlling the spread of CRAb in our ICU. This effect was maintained even during the first year of the COVID-19 pandemic and until Q4, 2020 after the Beirut blast, despite tremendous strain on our healthcare system [75,76,77,78]. The devastating explosion of August 2020 in Beirut caused an influx of trauma patients and was followed by a COVID-19 surge in the last quarter of 2020 leading to an increase in critically ill patients and antibiotic overuse [79]. 

Our results are particularly encouraging as reports are emerging regarding the potential worsening of antimicrobial resistance following the COVID-19 pandemic [17]. The impact of antimicrobial resistance in the countries of the East Mediterranean region is expected to worsen as well following the COVID-19 pandemic [80]. In addition, our hospital witnessed for the first time an outbreak of *C. auris* during the COVID-19 surge [81]. The lessons learnt during the multiple clusters and outbreaks of CRAb proved successful in controlling the spread of this new pathogen. In addition, the antimicrobial stewardship team adopted elements of antifungal stewardship in an effort to control the *C. auris* outbreaks.

Our study has limitations. First, we did not design our protocol to account for the impact of individual interventions on the outcomes. Thus, our approach was to maintain the infection control interventions and in parallel deploy the antimicrobial stewardship-targeted strategies as an additional combined intervention; ongoing infection control measures had not been fully effective in significantly reducing the colonization pressure of CRAb previously and we assumed that the intensification of the antimicrobial stewardship interventions resulted in the achieved reduction of CRAb rates and colonization pressure. Second, we did not collect data to study the impact of our results on patient outcomes such as cost, *C. difficile* infection, length of stay, and mortality.

## 5. Conclusions

In conclusion, we have shown a drastic reduction in CRAb colonization in our ICU and decreased resistance rates among *Acinetobacter baumannii* following a combination approach that relied on rigorous infection control practices and antimicrobial stewardship interventions. In our setting, the results are encouraging and could be replicated in hospitals and ICUs suffering from high burdens of CRAb transmission.

It is imperative to build on local experiences in comparable settings to develop successful protocols and implement adapted policies.

## Figures and Tables

**Figure 1 antibiotics-11-00911-f001:**
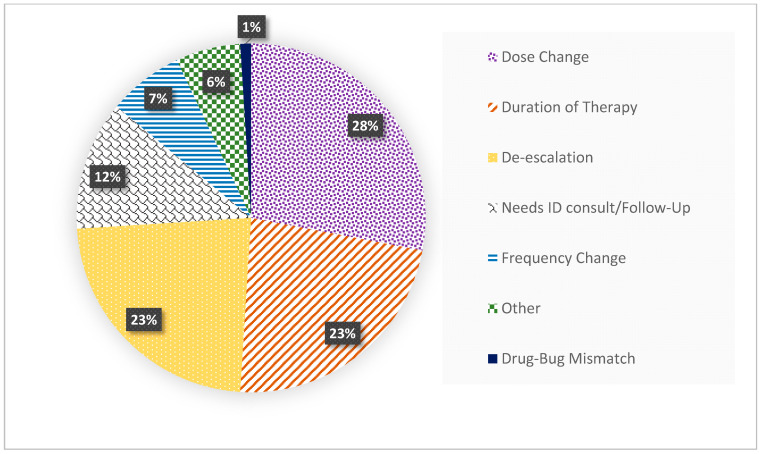
Distribution of antimicrobial stewardship interventions (n = 81) for patients receiving carbapenems during April 2019. ID, Infectious Disease.

**Figure 2 antibiotics-11-00911-f002:**
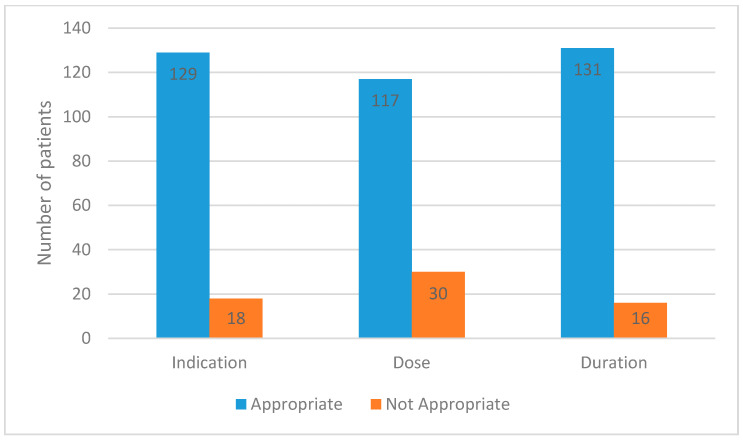
Appropriateness of carbapenem therapy per antimicrobial stewardship team during the implementation of carbapenem sparing strategies (April 2019).

**Figure 3 antibiotics-11-00911-f003:**
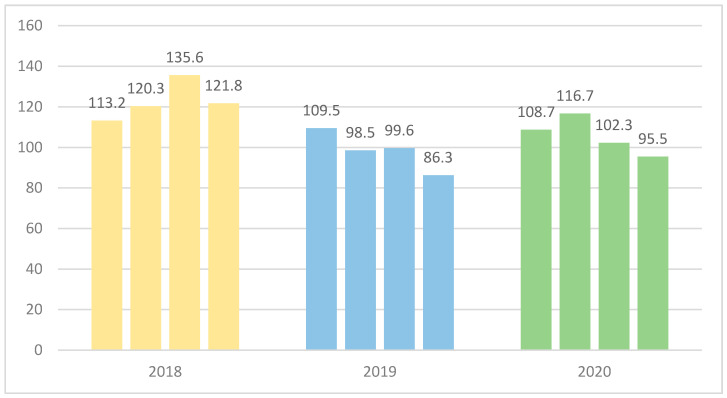
Carbapenems Defined Daily Dose per 1000 Patient Days per quarter and year.

**Figure 4 antibiotics-11-00911-f004:**
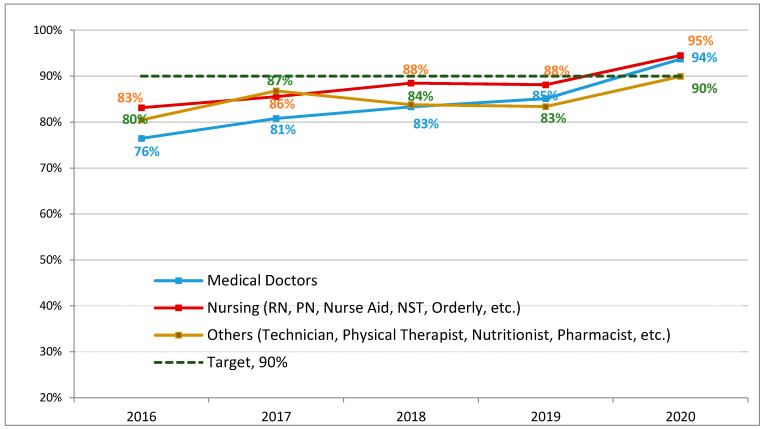
Hand hygiene compliance rates 2016–2020 based on anonymous audits. RN, registered nurse; PN, practical nurse; NST, nurse technician.

**Figure 5 antibiotics-11-00911-f005:**
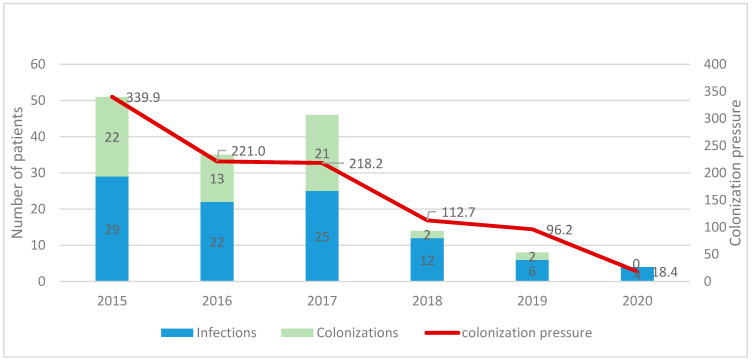
CRAb colonization pressure in ICU over a 7-year period by year.

**Figure 6 antibiotics-11-00911-f006:**
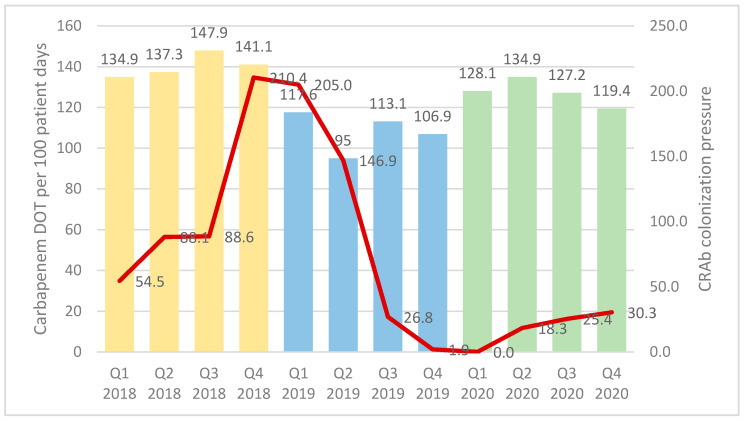
Carbapenem-resistant *Acinetobacter*
*baumannii* colonizing pressure and carbapenem consumption by quarter from 2018 until 2020. DOT, Days of therapy; Q, quarter.

**Figure 7 antibiotics-11-00911-f007:**
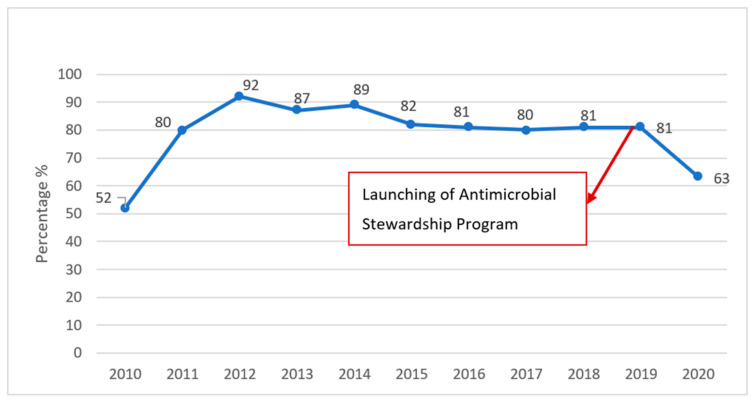
Rates of carbapenem-resistant *Acinetobacter baumannii* over years.

**Table 1 antibiotics-11-00911-t001:** Equations for Antibiotic Consumption Metrics and colonization pressure DDD, defined daily dose; DOT, days of therapy; CP, colonization pressure.

Metrics	Equations
DDD per 1000 patient days	$\frac{\sum dispensed\;doses\;of\;meropenem,\;ertapenem,\;imipenem\;}{1000\;\times \;patient\;days\;in\;a\;quarter}$
DOT per 1000 patient days	$\frac{\sum days\;that\;inpatients\;received\;\geq 1\;dose\;of\;meropenem,\;ertapenem,\;and\;imipenem}{1000\;\times \;patient\;days\;in\;a\;month}$ Days on which patients received more than one carbapenem are counted only once
CP	$\frac{\sum \;MDR\;-\;Ab\;patient\;days\;in\;a\;given\;unit\;in\;a\;month}{1000\;\times \;patient\;days\;in\;a\;month\;in\;same\;unit}$

## Data Availability

The data presented in this study are available on request from the corresponding author.

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
