# Peer review of "The Impact of Antimicrobial Stewardship and Infection Control Interventions on Acinetobacter baumannii Resistance Rates in the ICU of a Tertiary Care Center in Lebanon"

_antibiotics, 2022, doi:10.3390/antibiotics11070911_

Round 1
Reviewer 1 Report
"Carbapenem resistant" should be more accurately written as "carbapenem-resistant".
Some explanation/elaboration of "EPIC" on line 119 is needed - is this an acronym? Likewise, EPIC is also mentioned in Table 1 without any explanaion given.
In Table 1, therapeutic drug monitoring was mentioned as one of the ASP interventions. Please elaborate on this because TDM is widely available for only a small fraction of antibiotics.
On line 166, polymyxin B was referred to as "colistin". However, "colistin" is actually polymyxin E.
Some "Acinetobacter" (e.g. on lines 162, 171, 178, etc.) are not displayed in italics.
On line 179, it was mentioned that disk diffusion was used for antibiotic susceptibility testing. However the current CLSI document M100 specifically states that polymyxin B and colistin susceptibility cannot be tested by disk diffusion.
In the main text (lines 179-180), Table 1 is supposed to display the entire antibiogram of Acinetobacter baumannii (including resistance to non-carbapenem drugs). However, the actual Table 1 only shows resistance to carbapenems.
It would be great to state the concentrations of the chemicals mentioned, particularly the [chlorhexidine] used in baths, the [alchohol] contained in handrubs and the [H2O2] used for air decon.
For carbapenem-sparing strategies to work, a low ESBL rate amongst Enterobacterales is needed since CPs are the mainstay antibiotic to treat infections cause by ESBL-producing Enterobacterales. Is this the case in AUBMC?
The discussion appears to be superficial/too general.
References appear to be quite recent, with several cited papers being published circa 2020.
Author Response
- "Carbapenem resistant" should be more accurately written as "carbapenem-resistant".
Thank you for your comment. This was noted and modified in the text as suggested.
- Some explanation/elaboration of "EPIC" on line 119 is needed - is this an acronym? Likewise, EPIC is also mentioned in Table 1 without any explanation given.
Thank you for your comment: we clarified in the text as suggested.
- In Table 1, therapeutic drug monitoring was mentioned as one of the ASP interventions. Please elaborate on this because TDM is widely available for only a small fraction of antibiotics. At our hospital, therapeutic drug monitoring is done for aminoglycosides (amikacin, gentamicin) and vancomycin. Since TDM is not relevant to our topic, we will omit the mention in the text. We will delete Table 1 based on reviewer comments.
- On line 166, polymyxin B was referred to as "colistin". However, "colistin" is actually polymyxin E.
Thank you for your comment; this was corrected in the text.
- Some "Acinetobacter" (e.g. on lines 162, 171, 178, etc.) are not displayed in italics.
Corrected in the text as suggested.
- On line 179, it was mentioned that disk diffusion was used for antibiotic susceptibility testing. However the current CLSI document M100 specifically states that polymyxin B and colistin susceptibility cannot be tested by disk diffusion.
Thank you for your comment. We confirmed with our microbiology laboratory: the method used for susceptibility testing for colistin is Disc diffusion in our hospital. Other methods of susceptibility testing are not available.
- In the main text (lines 179-180), Table 1 is supposed to display the entire antibiogram of Acinetobacter baumannii (including resistance to non-carbapenem drugs). However, the actual Table 1 only shows resistance to carbapenems.
Thank you for your comment relevant to Figure 1. We corrected the caption of Figure 1 so that it only shows carbapenem resistant Acinetobacter baumannii and the rates (%). The order of the tables and figures was modified according to the comments received from the reviewers. This figure now is labeled Figure 7.
- It would be great to state the concentrations of the chemicals mentioned, particularly the [chlorhexidine] used in baths, the [alcohol] contained in handrubs and the [H2O2] used for air decon.
Thank you for the comment. We added the concentrations in the text and highlighted the changes, line 316, and line 324. Concentrations of chlorhexidine is 4%, that of alcohol is 70% ethanol or propanol, and H2O2 at a concentration of 1%, generating 4.7% boosted H2O2.
- For carbapenem-sparing strategies to work, a low ESBL rate amongst Enterobacterales is needed since CPs are the mainstay antibiotic to treat infections cause by ESBL-producing Enterobacterales. Is this the case in AUBMC?
Thank you for the comment. This is not the case at AUBMC as the ESBL rates are high (around 40% among Enterobacterales) However, even with carbapenems being the mainstay of therapy for ESBL organisms, recent data and guidance suggests using alternatives to carbapenems in several scenarios (intra-abdominal infections, complicated UTI and pyelonephritis, oral step-down therapy, surgical prophylaxis, etc.) to try to limit carbapenem use. One main aspect of our carbapenem-sparing strategy implemented was education of healthcare providers and distinguishing when it was possible to replace carbapenems with reasonable and efficacious alternatives without compromising patient care and safety.
- The discussion appears to be superficial/too general.
We have revised the discussion and rewrote some sections as recommended.
- References appear to be quite recent, with several cited papers being published circa 2020.
Thank you for this comment, the recent references used in the discussion aim at supporting our conclusion given the concern about rising resistance rates following the COVID-19 pandemic. Even though we conducted our study in 2018-2020, the implications and results appear to be very relevant today.
Reviewer 2 Report
This manuscript fitted well in the scope of journals with a issue on antimicrobial resistance. The timely update of AMR in bacterial pathogen is necessary and important. Authors nicely addressed that how antibiotic stewardship, infection control, good laboratory practices impacted on the bacterial colonization and antibiotic resistance in hospital. However, there are some limitations of the study conducted and needs some modification and correction before its publication. The manuscript is written well and need a moderate changes.
In my opinion manuscript might be accepted for publication provided that improvement in text required.
Major comments:
Introduction part needs to improved, authors only talk about the A. baumannii antimicrobial resistance. Please talk about its excellent genome plasticity i.e. ability take any gene from surroundings, this feature might have played a crucial role in the evolution of this human opportunistic pathogen towards clinical success and being a MDR pathogen. It has island of drug resistant gene in its genome that makes it different from other superbugs. Talk about this part line No. 84 onwards.
Line 76 : Following covid-19 pandemic, AMR rates are expected to increase. This is not totally true, although COVID-19 makes life harder for those patient in ICU and had to take antibiotics. However, in broader picture COVID-19 has minimized the spread of these superbugs due to use of face mask, frequent hand sanitization and restriction in global travels. Please discuss it as well.
The conclusion is well supported by the results, although some limitations are there which already mentioned by the authors in the manuscript.
Minor comments
Please write Clostridium difficile once and afterwards change to "C. difficile" elsewhere, please apply same for other organism in manuscript.
Author Response
This manuscript fitted well in the scope of journals with a issue on antimicrobial resistance. The timely update of AMR in bacterial pathogen is necessary and important. Authors nicely addressed that how antibiotic stewardship, infection control, good laboratory practices impacted on the bacterial colonization and antibiotic resistance in hospital. However, there are some limitations of the study conducted and needs some modification and correction before its publication. The manuscript is written well and need a moderate changes.
In my opinion manuscript might be accepted for publication provided that improvement in text required.
Major comments:
- Introduction part needs to improved, authors only talk about the A. baumannii antimicrobial resistance. Please talk about its excellent genome plasticity i.e. ability take any gene from surroundings, this feature might have played a crucial role in the evolution of this human opportunistic pathogen towards clinical success and being a MDR pathogen. It has island of drug resistant gene in its genome that makes it different from other superbugs. Talk about this part line No. 84 onwards.
Thank you for your comment. We have added more details supported by important references about the mechanisms of resistance of Acinetobacter baumannii as suggested.
- Line 76 : Following covid-19 pandemic, AMR rates are expected to increase. This is not totally true, although COVID-19 makes life harder for those patient in ICU and had to take antibiotics. However, in broader picture COVID-19 has minimized the spread of these superbugs due to use of face mask, frequent hand sanitization and restriction in global travels. Please discuss it as well.
Thank you for your comment. multiple sources such as the WHO and CDC warn about the potential increase in antimicrobial resistance following the COVId-19 pandemic. Data presented at various international conferences suggest this trend, awaiting publication of results. However, your comment is well taken and we added a statement about possible decrease in transmission of resistant organisms as a direct consequence of global travel restrictions, more frequent hand hygiene, social distancing, as well as enhanced infection control practices globally.
- The conclusion is well supported by the results, although some limitations are there which already mentioned by the authors in the manuscript.
Minor comments
- Please write Clostridium difficile once and afterwards change to "C. difficile" elsewhere, please apply same for other organism in manuscript.
Thank you for your comment. Corrected in the text as suggested.
Reviewer 3 Report
The authors did a nice job, where an important topic is covered! While reading some points need to be revised:
- Starting from the title, the referee suggests improving and correcting it, because it is very confusing and long; furthermore, the dot at the end of the title must be deleted (lines 2-5).
- Please respect the same acronym after the long name of the respective bacteria used first. In line 80, italicize the bacterium acronym.
- In line 178 you must first mention the long name of MALDI-TOF and then use its acronym.
- The titles of figures 1-8 must be moved to the bottom of the graph, not as a header as is used for tables. It is necessary to check their titles again, because there are the full names of the acronyms used before, a useless repetition of the terms.
- In Figure 1, in the title of the graph, enter the acronym of A. baumannii and delete the AUBMC in line 185, because it is mentioned above.
- In the table 3, instead of “inpatients”, the referee suggests to use “hospitalized”. Delete DDD in line 197.
- In Figure 5, use the unit of the “daily dose” used for the patient, within the graph. The same for Figure 4; the title of it is unclear.
- Please verify for the repetition of the long name of CP, after it is used for the first time. In the abstract is used twice.
- Delete repeated full names and their acronyms in lines 252 and 264.
- Put the acronym of C. auris in line 388 and of C. difficile in line 398.
- On line 421 enter the name “References” and adjust the references below according to the guidelines of the journal.
Author Response
Review 3
Comments and Suggestions for Authors
The authors did a nice job, where an important topic is covered! While reading some points need to be revised:
- Starting from the title, the referee suggests improving and correcting it, because it is very confusing and long; furthermore, the dot at the end of the title must be deleted (lines 2-5).
Thank you for your comment. We modified the title as suggested.
- Please respect the same acronym after the long name of the respective bacteria used first. In line 80, italicize the bacterium acronym.
Thank you for your comment. Corrected in the text as suggested.
- In line 178 you must first mention the long name of MALDI-TOF and then use its acronym.
Thank you for your comment. Corrected in the text as suggested.
- The titles of figures 1-8 must be moved to the bottom of the graph, not as a header as is used for tables. It is necessary to check their titles again, because there are the full names of the acronyms used before, a useless repetition of the terms.
Thank you for your comment. The tables and their titles were modified based on the feedback received from the reviewers and following the guidelines of the journal.
- In Figure 1, in the title of the graph, enter the acronym of baumanniiand delete the AUBMC in line 185, because it is mentioned above.
Thank you for your comment. Corrected in the caption as suggested. Some of the tables and figures were modified and others deleted as suggested by other reviewers.
- In the table 3, instead of “inpatients”, the referee suggests to use “hospitalized”. Delete DDD in line 197.
Thank you for your comment. Corrected in the text as suggested.
- In Figure 5, use the unit of the “daily dose” used for the patient, within the graph. The same for Figure 4; the title of it is unclear.
We corrected and modified the title of figures as suggested.
- Please verify for the repetition of the long name of CP, after it is used for the first time. In the abstract is used twice.
Thank you for your comment. We omitted several abbreviations as suggested by the reviewers and because CP can be mistaken for carbapenem.
- Delete repeated full names and their acronyms in lines 252 and 264.
Thank you for your comment. Corrected in the text as suggested.
- Put the acronym of aurisin line 388 and of C. difficile in line 398.
- Thank you for your comment. Corrected in the text as suggested.
- On line 421 enter the name “References” and adjust the references below according to the guidelines of the journal.
Thank you for your comment. Corrected in the text as suggested.
Reviewer 4 Report
The manuscript Rizk et al. describes antimicrobial stewardship and infection control interventions due to the high prevalence of carbapenem-resistant strains of Acinetobacter baumannii in the intensive care unit. The authors report a significant decrease in colonization pressure reaching record low levels after the introduction of antimicrobial stewardship and infection control measures. In connection with the alarming situation regarding antibiotic resistance, the subject is very interesting for the reader, however the manuscript needs to be rewritten and restructured for better understanding and readability. Here are some comments, which should improve clarity of the manuscript:
- There are a lot of abbreviations in the text which, due to their excessive amount, impair fluidity and understanding, e.g. there are 10 in the abstract alone. Please leave in full: antimicrobial resistance, antimicrobial stewardship, antimicrobial stewardship program, infection control, colonization pressure, carbapenem, Eastern Mediterranean region, global action plan against antimicrobial resistance, multidrug resistant organisms, Center for Disease Control.
- Line 119: on the contrary, explain the acronym EPIC
- Line 130: list specific objectives of the program
- Delete tables 1 and 2 and formulate their contents in the text
- Section Materials and Methods needs to be rewritten so that the reader understands the specific methods used during the study in your hospital. General methods should be included in the introduction.
- define patient population, how many, their diagnosis, demographic data
- define time period of the study
- describe microbiological examinations including sampling: what biological materials are examined at admissions of patients, how many times per week, what methods are used for identifications of bacterial strains, determination of susceptibility to antibiotics and detection of carbapenemases
- describe exactly and chronologically what infection control and antimicrobial stewardship interventions were implemented, how and when they were checked and evaluated, when they ended or changed
- define, how the consumption of carbapenems was assessed
- describe, how the identification of each cluster or outbreak was carried out
6) Section Results:
- how many patients were included in each year of the study, how many Acinetobacter baumannii strains per year, how many strains (%) were carbapenem-resistant, how many strains were carbapenemase producers, how many strains caused infection and how many were only commensals
- consumption of carbapenems: Figure 4 and 5 join to one chart, define specific carbapenem dose regimens, mean duration of therapy, proportion of monotherapy and combination therapy (it would be interesting to know, whether the regimens changed significantly after the ASP interventions)
- please correlate prevalence of resistant strains in individual years with carbapenem consumption
- Figure 3: compare periods before and after ASP introduction
- Figure 8 is redundant, CP is in Fig 7 and consumption of carbapenems is in Fig 4
7) Lines 163-176 formulate in the Discussion section.
8) Lines 383-386 are not related to the topic, please remove it.
Author Response
Comments and Suggestions for Authors
The manuscript Rizk et al. describes antimicrobial stewardship and infection control interventions due to the high prevalence of carbapenem-resistant strains of Acinetobacter baumannii in the intensive care unit. The authors report a significant decrease in colonization pressure reaching record low levels after the introduction of antimicrobial stewardship and infection control measures. In connection with the alarming situation regarding antibiotic resistance, the subject is very interesting for the reader, however the manuscript needs to be rewritten and restructured for better understanding and readability.
Here are some comments, which should improve clarity of the manuscript:
- There are a lot of abbreviations in the text which, due to their excessive amount, impair fluidity and understanding, e.g. there are 10 in the abstract alone. Please leave in full: antimicrobial resistance, antimicrobial stewardship, antimicrobial stewardship program, infection control, colonization pressure, carbapenem, Eastern Mediterranean region, global action plan against antimicrobial resistance, multidrug resistant organisms, Center for Disease Control.
Thank you for your comment. Those abbreviations were all deleted and modified based on your suggestion.
- Line 119: on the contrary, explain the acronym EPIC
Thank you for your comment. The EPIC system is an electronic medical record, this was clarified in the text as suggested.
- Line 130: list specific objectives of the program
The objectives of the program were listed in the text as suggested.
- Delete tables 1 and 2 and formulate their contents in the text
Thank you for your comment. This suggestion was carried out and we deleted tables 1 and 2.
- Section Materials and Methods needs to be rewritten so that the reader understands the specific methods used during the study in your hospital. General methods should be included in the introduction.
Thank you for your detailed comments and suggestions. Those were all addressed in the Methods section and the section was re-written based on your input.
- define patient population, how many, their diagnosis, demographic data clarified. We did not collect patient information, only data on the number and types of interventions.
- define time period of the study done
- describe microbiological examinations including sampling: what biological materials are examined at admissions of patients, how many times per week, what methods are used for identifications of bacterial strains, determination of susceptibility to antibiotics and detection of carbapenemases done
- describe exactly and chronologically what infection control and antimicrobial stewardship interventions were implemented, how and when they were checked and evaluated, when they ended or changed clarified
- define, how the consumption of carbapenems was assessed done
- describe, how the identification of each cluster or outbreak was carried out
We described the infection control interventions without mentioning the details of outbreak investigations, which were reported previously and added the pertinent references.
6) Section Results:
- how many patients were included in each year of the study, how many Acinetobacter baumannii strains per year, how many strains (%) were carbapenem-resistant, how many strains were carbapenemase producers, how many strains caused infection and how many were only commensals
Thank you for this pertinent comment. We added details to clarify this point however some of the information you inquire about is not available to us. We collected information on the number of interventions and do not have the exact number of ICU patients whose records were reviewed daily.
Our diagnostic microbiology laboratory does not perform routine molecular analysis on all isolated A. baumannii samples. This can be performed for research purposes by our Molecular laboratory.
- consumption of carbapenems: Figure 4 and 5 join to one chart, define specific carbapenem dose regimens, mean duration of therapy, proportion of monotherapy and combination therapy (it would be interesting to know, whether the regimens changed significantly after the ASP interventions)
Thank you for your comment. We removed the figure showing DOT and kept the DOT data along with CRAb colonization pressure in a separate figure (Figure 6) to avoid redundancy. DDD data is kept as Figure 3. We agree that it would be enriching to include data about specific carbapenem therapy such as dose regimens, mean duration of therapy, proportion of monotherapy and combination therapy, we unfortunately did not collect any of this information for the entire period before and after the intervention and are currently in lack of it. We however have data on the mean duration of therapy of carbapenems during the month of April 2019, and that is an average of 5 +/- 4 days. We also have data on the mean duration of therapy in the months of January 2019 and 2020 (pre- and post-intervention) of 4.8 days and 4.7 days, respectively.
Regarding the appropriateness of therapy: when comparing the months of January 2019 and January 2020 pre- and post-intervention period in another study, the indication for use in empirical therapy before 48 hours was improved from 86.4% to 92.9%. Similarly, indication for use in empirical therapy after 48 hours from culture results, and indication for targeted therapy were appropriate in 67.1% (January 2019) and 78.9% (January 2020), and 88.9% (January 2019) and 91.4% (January 2020) of the cases respectively. Duration and dosing regimen were appropriate in 64.3% and 75.8% in January 2019 respectively as opposed to appropriateness rates of 72.1% and 68.7% in January 2020.
- please correlate prevalence of resistant strains in individual years with carbapenem consumption
Thank you, this information was added to the text and we modified the CRAb resistance figure to add more details. The carbapenem consumption before 2018 has not been calculated for the years prior.
- Figure 3: compare periods before and after ASP introduction
Thank you, this was clarified. As a result of all antimicrobial stewardship efforts, for 2019, there was an increase in stewardship interventions acceptance rate (16.66 % vs 55.95%) comparing Jan 2019 to Jan 2020 (p=0.03) considering that even though the antimicrobial stewardship team was active, however the efforts were less focused and spanned over the whole hospital (vs. April 2019 when the efforts were focused on the ICUs).
- Figure 8 is redundant, CP is in Fig 7 and consumption of carbapenems is in Fig 4
We deleted that figure as suggested and modified the order of the figures after re-writing the Methods and Results sections.
7) Lines 163-176 formulate in the Discussion section.
Thank you for your comment. We believe that keeping this point in this section to describe Acinetobacter resistance fits with our objectives and we have alluded again to this point in the discussion as recommended.
8) Lines 383-386 are not related to the topic, please remove it.
Thank you for your comment and suggestion. We have edited the Discussion section based on feedback received. in our conclusion, we highlight the struggles in controlling antimicrobial resistance in our region and the need to develop adapted policies. In this context, we set to briefly describe the reality and present our findings (reducing colonization pressure and resistance rates among Acinetobacter spp at our hospital) from this perspective.

Round 2
Reviewer 4 Report
After editing, the revised manuscript of Rizk et al. is much better, clearer and more understandable, however there are a few minor points that should be corrected:
- Paragraph 2.2 - this paragraph by its nature does not belong to the methodology. It can be placed (and reformulated) in the Introduction section or in the Discussion section or divided into the introduction (general facts) and the discussion (facts related to AUBMC).
- Line 149 - Bactrim is a company name, please replace it with a chemical name
- Line 150 - remove the dot between the brackets
- Line 243 - replace IC with the full word “infection control”
- Figure 1 - there are 7 slices of the graph, but there are only 5 points in the graph label, you need to correct it. In black and white version, the graph is badly readable, consider another variant of the graph, or conversion to text. Legend of the graph (explanation of abbreviations) is redundant, delete it.
- Line 299 - I do not understand the percentage 88%, 80%, 89%. In the text above, the authors state that 46% of carbapenem prescriptions were inappropriate, in the line 299 is stated that 88%, 80%, 89% of the indications for use, dosing and duration were appropriate. Please explain this in more detail.
- Lines 300 - 307 describe the results of another study (reference 55), which does not belong into the Result section but into the Discussion section. Please relocate.
- Figure 2 Line 312 - please write the acronym ASP in full. Specify, which time period. April 2019? Or longer?
- Figure 2 - describe the Y axis (Number of patients). In the chart label write only Appropriate and Not appropriate (delete N and %). Delete the legend (line 316), it is redundant.
- Line 336 - 339: The results of another study (reference 62) belong to the Discussion section.
- Line 339 - 340: elaborate more, e.g. it is interesting, that doctors had the lowest hand hygiene compliance rate in 2016 and nurses the highest one, while in 2020 the values reached the same level.
- Figure 5: describe the Y axis (Number of patients)
- Figure 6: Chart label (CBP DOT per 1000 patients days) put as Y axis label
- Line 381: 925 is not correct, is it 92%?
- Line 384: when you say “significantly”, please add p-value
- Figure 7 - describe the Y axis (%)
- In the whole text: unify the use of the name Acinetobacter baumannii. Somewhere you use A. baumannii, somewhere else you use Acinetobacter spp. E.g. Line 378 you write Acinetobacter spp. while refering to Figure 7. In the name of Figure 7 there is Acinetobacter baumannii. Please unify the name of the species and the form (full name) throughout the text.
Author Response
We appreciate the time and input of the reviewers and editor. We found those comments and suggestions to be pertinent and constructive.
Please find here below a point-by-point response to the reviewer’s comments and suggestions. We have edited our manuscript and figures/graphs accordingly.
- Paragraph 2.2 - this paragraph by its nature does not belong to the methodology. It can be placed (and reformulated) in the Introduction section or in the Discussion section or divided into the introduction (general facts) and the discussion (facts related to AUBMC).
Thank you for your comment and suggestion. We reformulated the paragraph and included the information in the Introduction section.
- Line 149 - Bactrim is a company name, please replace it with a chemical name
Many thanks again for your thorough review, this was corrected accordingly.
- Line 150 - remove the dot between the brackets
This dot was removed, thank you.
- Line 243 - replace IC with the full word “infection control”
This was replaced as suggested.
- Figure 1 - there are 7 slices of the graph, but there are only 5 points in the graph label, you need to correct it. In black and white version, the graph is badly readable, consider another variant of the graph, or conversion to text. Legend of the graph (explanation of abbreviations) is redundant, delete it.
Thank you again for the observation, the legend was edited as suggested; the 7 points that correspond to the seven slices of the graph are now clearer. We revised the graph in such a way to allow easier readability in black and white.
- Line 299 - I do not understand the percentage 88%, 80%, 89%. In the text above, the authors state that 46% of carbapenem prescriptions were inappropriate, in the line 299 is stated that 88%, 80%, 89% of the indications for use, dosing and duration were appropriate. Please explain this in more detail.
Thank you again for your comment. During this study period, the antimicrobial stewardship team recommended to stop or de-escalate from carbapenem use in 46% of carbapenem prescriptions (recommendation to de-escalate (23%) or stop therapy (23%)). Those recommendations were labeled as “stewardship interventions”.
In analyzing the indication for use of carbapenems by the antimicrobial stewardship team, we defined appropriate empiric therapy with carbapenem as follows: patient is a candidate for broad antibiotic therapy and warrants carbapenem usage such as recent culture with ESBL Enterobacterales or other multi-drug resistant organisms, sepsis, or febrile neutropenia. Therefore 88% of prescriptions were appropriate initially and may require adjustment subsequently.
- Lines 300 - 307 describe the results of another study (reference 55), which does not belong into the Result section but into the Discussion section. Please relocate.
Thank you for your comment. We agree with your suggestion and we removed this reference and relocated it as suggested.
- Figure 2 Line 312 - please write the acronym ASP in full. Specify, which time period. April 2019? Or longer?
Thank you for this observation, we edited the figure and caption and legend as suggested.
- Figure 2 - describe the Y axis (Number of patients). In the chart label write only Appropriate and Not appropriate (delete N and %). Delete the legend (line 316), it is redundant.
We edited the figure and caption and legend as suggested.
- Line 336 - 339: The results of another study (reference 62) belong to the Discussion section.
Thank you for your comment. We relocated this point as suggested to the Discussion part.
- Line 339 - 340: elaborate more, e.g. it is interesting, that doctors had the lowest hand hygiene compliance rate in 2016 and nurses the highest one, while in 2020 the values reached the same level.
Thank you for your comment. We further clarified this point in the text (Methods section) and added the appropriate reference: Tiered hand hygiene accountability interventions were adopted based upon a validated model and was reflected in the hospital hand hygiene policy. Interventions started with direct feedback followed by the awareness intervention, then the authority intervention and ending with the disciplinary intervention. Hand hygiene compliance rates started to improve for the physician group as a result. Historically at our institution, the compliance with hand hygiene among physicians was lower than other groups of healthcare workers. We intensified the training and awareness targeting physicians. Hand hygiene compliance rates were sustained and improved further at the start of the COVID-19 pandemic.
- Figure 5: describe the Y axis (Number of patients)
The figure was edited as suggested, thank you.
- Figure 6: Chart label (CBP DOT per 1000 patients days) put as Y axis label
The figure was edited as suggested, thank you.
- Line 381: 925 is not correct, is it 92%?
Thank you for this pertinent observation, this was corrected to 92% as you suspected.
- Line 384: when you say “significantly”, please add p-value
Thank you for your comment. We removed the word “significantly” because we do not have a p-value calculated.
- Figure 7 - describe the Y axis (%)
This was edited as you suggest.
- In the whole text: unify the use of the name Acinetobacter baumannii. Somewhere you use baumannii, somewhere else you use Acinetobacterspp. E.g. Line 378 you write Acinetobacter spp. while referring to Figure 7. In the name of Figure 7 there is Acinetobacter baumannii. Please unify the name of the species and the form (full name) throughout the text.
Thank you for this pertinent comment. During the study period, all resistant Acinetobacter species belonged to the group of Acinetobacter baumannii. Acinetobacter baumannii constitute the large majority of the Acinetobacter organisms tested in our microbiology diagnostic laboratory. For the purpose of this study, we will use the term Acinetobacter baumannii to refer to all Acinetobacter organisms.